# Job Perceptions Contribute to Stress among Secondary School Teachers in Southwestern Uganda

**DOI:** 10.3390/ijerph18052315

**Published:** 2021-02-26

**Authors:** Joseph Ssenyonga, Tobias Hecker

**Affiliations:** 1Department of Psychology, University of Konstanz, 78567 Konstanz, Germany; joseph.ssenyonga@uni-konstanz.de; 2Department of Psychology, University of Bielefeld, 33501 Bielefeld, Germany

**Keywords:** teacher, stress, working conditions, feeling of pressure at work, teaching difficulties

## Abstract

(1) Background: Teachers’ personal and strenuous working conditions reflect the realities of the teaching vocation that may result in increased stress levels and associated negative consequences, such as negative emotions. It is also well-known that teacher stress contributes to more violence against students. However, little is known about personal and school context factors that contribute to teachers’ stress. The current study examined whether, in addition to school-related factors, job perceptions, including the feeling of pressure at work and perceived school climate and teaching difficulties, contribute to teachers’ stress. (2) Methods: A representative sample of 291 teachers from 12 public secondary schools in southwestern Uganda responded to self-administered questionnaires. (3) Results: Teaching difficulties and feelings of pressure at work contributed to teachers’ stress. Furthermore, stress did not vary with teachers’ sociodemographic variables. (4) Conclusions: Teachers’ perceptions of their working conditions were associated with teacher stress levels. Therefore, more efforts need to be geared towards improving the working conditions of teachers as a way of reducing stress.

## 1. Introduction

Teaching is a highly stressful profession, and the characteristics of teachers’ work seem to be related to stress and associated negative emotions [1]. The teaching profession is highly regulated, which limits the individual scope for decision-making, e.g., teachers have to conform to the Ministry of Education guidelines and teacher professional code of conduct [2]. In addition, teacher stress may be influenced on different levels, for example, on the individual level (e.g., teachers feel isolated due to a lack of mentorship or inadequate support by professional networks), the interpersonal level (e.g., interacting with demanding parents), or the organizational level (e.g., policy issues may result in teacher strain) [3].

In low-income countries, teachers are further burdened with often poor working conditions, and these terms and conditions of service may result in stress. For example, teachers interact and work with students daily, but, in addition, they have to take over multiple other tasks and roles in the school, such as teaching, being a class teacher, being a sports or career guidance master, and being the head of a department, that altogether increases their workload [4]. More so, the low-paid teachers receive little recognition for their work from society [5]. Furthermore, teaching is a routinized job that involves imparting the same material with insignificant challenge and diversity. Teachers in low- and middle-income countries also often report that they do not have an interest in the teaching vocation because of a lack of career advancement opportunities and poor remuneration, and that they are frustrated by working in a limited resource environment [4,5,6]. In addition, poor education of teachers in low-income countries may also explain the reported feelings of helplessness and that teachers are overstrained when dealing with misbehaving students [7]. Often teachers in low-income countries have only limited access to school resources but have to educate large classes for long hours and for meager remuneration [8] that may not be able to cater to the needs of their families and other dependents [5]. More so, teachers work with little support from their supervisors and school management, despite the obvious challenges. Often supervisor feedback about teachers’ performance is negative, accompanied by unwelcome criticism, yet aimed at finding only faults [4].

A nexus of teacher personal attributes, such as age, as well as contextual and environmental factors, including school and class size [9], may altogether result in heightened teacher stress levels. In essence, teachers may experience stress and negative emotions whenever they cannot adequately fulfill their duties as teachers [1]. Likewise, teacher daily routines, for example maintaining student discipline, time pressures, a large class size, amplified workload, role conflict, and poor working conditions, are associated with teachers’ stress [1,10,11]. Furthermore, teachers with under three years of teaching practice experience stress when dealing with demanding or challenging students and due to organizational aspects, which result in negative emotions that further affect their teaching behavior, including efficient classroom management and providing an appropriate learning environment [12]. Therefore, understanding the interplay between work-related aspects and personal factors may help us to determine factors contributing to teachers’ stress.

Teacher stress has negative consequences for the individual teachers, such as mental health and psychological problems [3], and the education sector, for instance, when teachers leave the demanding teaching profession [13]. In addition, teachers’ stress is associated with undesirable aftereffects, including poor teacher and student relations, neglecting students in the decision-making process, and a lack of care shown by teachers for their students [14]. Increased stress can result in conflict with colleagues, students, and parents, classroom management problems, isolation, and self-doubt [3]. Overall, teachers’ stress levels increase the likelihood of employing diverse violent techniques with varying intensities to control students’ behavior during school time [8,15]. However, the legitimate use of violent behaviors in schools implies that its use will be instantaneous in response to identified and perceived students’ misbehavior. By using violence, teachers are motivated by the desire to assert their authority, control students’ behavior, instill respect and obedience, deter future misbehavior, and ensure that children behave in accordance with sociocultural expectations [16,17,18]. All of these explanations suggest that violence by teachers is used specifically to forbid misconduct. The assumption is that the misconduct determines the punishment and thus the extent of violence. However, this is far from being the case: studies with teachers and students clearly show that the current mood, current strain, and perceived stress play an important role in determining the extent of violence by teachers [8,15]. This means that teachers’ stress is associated with various negative educational consequences. Therefore, there is a need to focus on factors that are likely to be changed in interventions, such as perceived stress and strain. It is essential to understand which factors contribute to the extent of teachers’ stress. However, so far, little is known about the factors that contribute to teachers’ stress in the high-strain school contexts of Sub-Saharan Africa. Therefore, there is a need to understand the dynamics of teacher stressors before designing, implementing, and evaluating interventions that target teacher behaviors, especially in societies in Sub-Saharan Africa where teacher-associated school violence is highly prevalent.

Teachers in Sub-Saharan Africa work in demanding and nerve-racking situations that are likely to increase their stress levels. For example, underpaid teachers experience difficulties maintaining discipline in large classes with a high teacher to student ratio [5]. Among Tanzanian teachers, stress was associated with working for long hours in emotionally exhausting situations [7]. In Uganda, teachers’ stress was attributed to an increase in students’ enrollment without the comparable corresponding recruitment of teachers. Hence, the available teachers experienced increased workloads and obligations on top of classroom management difficulties [6]. Teachers’ dissatisfaction with and perceptions of their terms and conditions of service may also contribute to stress. Feelings of pressure in the work context and teaching problems affected their satisfaction with the teaching vocation. Moreover, age, class size, and socioeconomic status were related to feelings of pressure and stress [8].

The levels of stress experienced by teachers also varied with individual attributes. Female teachers [19,20] who were married and who cared for younger children have shown an increased risk of stress. Furthermore, age [8,20,21,22] and teaching experience [20] were also associated with increased demands and stress. For example, secondary school teachers who have a bachelor’s level of education, a highly demanding job, less than five years of teaching experience, and poor relations with other staff experienced increased stress [23].

As increased stress levels are associated with negative outcomes, stress management techniques have been suggested as an essential part of teacher training efforts [15]. Therefore, there is a need to understand what factors contribute to stress and how these can be mitigated or amended as part of prevention efforts. Understanding these factors will be vital in designing interventions that will enable teachers to manage stress effectively, and to suggest structural changes to improve teachers’ working conditions in the long-term.

### Objectives

Teaching is considered as a stressful vocation [1,3,4,8,15] that affects individual teachers differently [19,20,21,22,23]. Teachers’ current stress and mood contribute to negative short- and long-term consequences, such as the application of violence in response to students’ misconduct. As teachers’ stress levels are the easiest amendable factors that may contribute to negative outcomes, it is of utmost importance to deepen our understandings of the personal and work-related factors that contribute to teachers’ stress. Therefore, we hypothesize that job perceptions, i.e., feelings of pressure at work and perceived school climate and teaching difficulties, contribute to teachers’ stress. Additionally, we predict that teacher stress levels will vary with sociodemographic and school-related variables.

## 2. Materials and Methods

### 2.1. Study Setting

The education system in Uganda takes on the 7-6-3 model, with seven years of primary education, six years of secondary education (four years of ordinary level or lower secondary and two years of advanced level or upper secondary), and 3–5 years of tertiary education. The study was conducted in the Ankole region of southwestern Uganda, with 3254 teachers (26.6% female) employed in the 103 government-aided secondary schools and with a student enrollment of 61,825 and an equal number of male and female students [13].

### 2.2. Design and Sampling

The study was conducted in six of the 10 districts of the Ankole region in southwestern Uganda from April to November 2017. The region had a total of 41 public secondary schools with a student enrollment of at least 360 and about 15 teachers. We randomly selected five districts and added one district with the largest population in the region. Then, from each district, we randomly selected two government-aided secondary schools, resulting in a sample of 12 schools. In total, all of the 435 teachers employed at these schools were eligible to participate in this study.

### 2.3. Procedure and Data Assessment

Ethical approval for the study was obtained from the Uganda National Council of Science and Technology (UNCST: SS 4032), the Research Ethics Committee of Mbarara University of Science and Technology (MUST REC 15/10-15), Uganda, and the Ethical Review Board of the University of Konstanz, Germany (35/2016). Before the start of the study, one Ugandan researcher (J.S.) and two trained research assistants visited all of the selected schools and discussed the relevant study details with the school administration and teachers as a way of eliciting their collaboration during the study. Before participating, teachers signed an informed consent document and then responded to the anonymous self-report questionnaire for about 20–30 min. In total, 291 of the 435 eligible teachers participated in the study (response rate: 67%).

### 2.4. Participants

The participating teachers (25.4% females) were in the age range between 23 and 59 years (M = 37.7, SD = 8.78) and had a mean teaching experience of 12.67 years (*SD* = 8.79, range: 0.42–43.75, median: 11 years). Most teachers (71.1%) were bachelor’s degree holders, who spent on average 42.69 h per week (*SD* = 30.49, range: 4–168) at the school, teaching an average of 68 students (*SD* = 46.70). Teachers (54.5%) had additional sources of income, including other jobs and owning businesses.

### 2.5. Measures

We first elicited teachers’ demographic information, including gender, age, academic qualifications, teaching experience, and class size.

#### 2.5.1. Teachers’ Perceptions of Their Work Conditions

Teachers’ attitudes, perceptions of their work conditions, and job satisfaction were assessed using an adapted and modified version of the Attitudes Toward Personal Teaching Behaviors Scale (ATPTB) [24] that has been used previously in Tanzania [8]. The modified ATPTB has two subscales, namely feelings of pressure in the work context (13 items; α = 0.73) and perceived classroom climate and teaching difficulties (14 items; α = 0.73). The items are scored on a Likert scale ranging from not true (0) to certainly true (2). Items that were positively phrased were recorded before data analysis. The sum of the responses to the respective items provided the subscale scores of feelings of pressure in the work context (range: 0–26) and perceived classroom climate and teaching difficulties (range: 0–28), with high scores affirming more problems with the teaching job. In the present study sample, the Cronbach alpha coefficients were 0.70 for the feeling of pressure in the work context and 0.73 for classroom climate and teaching difficulties.

#### 2.5.2. Stress and Burn Out

The 19-item Copenhagen Burnout Inventory (CBI) with three subscales that are work-related (α = 0.87), client-related (α = 0.85), and personal burnout (α = 0.87) assessed teacher’s stress levels and burnout [25]. Items are scored on a five-point Likert scale from never (0) to always (100). The averages of the respective items provide the CBI total and subscale scores, with scores > 25 indicating elevated stress levels and scores ≥ 50 high-stress levels [26]. The Cronbach alpha coefficient for the current study was 0.90 for the total score, 0.83 for student-related, 0.74 for work-related, and 0.82 for personal burnout.

### 2.6. Data Analysis

Pearson correlation coefficients were used to determine associations among the study variables. Regression analysis was used to examine the predictors of stress among teachers. All requirements for a multiple regression analysis were fulfilled. The independent variables were not a combination of other variables, hence satisfying the assumption of singularity. We did not need to take multicollinearity into account, as the tolerance was less than one, and the variance inflation factor did not exceed 1.15. Neither univariate nor multivariate outliers were detected. The plots of standardized residuals and scatter plots revealed that the data satisfied the assumptions of normality, linearity, and homoscedasticity. Data were analyzed with IBM SPSS statistics 26 (IBM Corp., Armonk, NY, USA).

## 3. Results

### 3.1. Teachers’ Burden and Satisfaction of Work

Teachers experienced feelings of pressure in their working context (*M* = 13.37, *SD* = 4.53, range: 3–22) that arose from several factors (see Table 1), including lack of public funding for the education sector (level of agreement: 90%), having trouble handling disruptive students in class (83.2%), and dealing with personal crises that interfere with life as a teacher (82.4%). Furthermore, a Mann–Whitney test indicated that the feeling of pressure in the working context was significantly greater for male teachers (median = 149.87) than for female teachers (median = 126.46, *U* = 6477, *p* = 0.038). On the other hand, teachers reported on the atmosphere at school and their perceived teaching difficulties regarding the teaching profession (*M* = 5.48, *SD* = 3.56, range: 0–25), for instance, not feeling competent in the subject taught (94%), becoming involved in the personal lives of their students (85.2%), and organizing the classroom for instruction (52.6%).

### 3.2. Teachers’ Stress and Burnout

The majority of teachers (60.2%) experienced elevated or high levels of stress (*M* = 30.46, *SD* = 15.85, range: 2.63–94.74) in diverse aspects of their work. Likewise, a majority of them (59%) reported work-related stress (*M* = 29.29, *SD* = 16.23, range: 0–89.29), while 51% of the teachers experienced student-related stress (*M* = 26.12, *SD* = 20.17, range: 0–100) (see Table 2). Furthermore, teachers’ stress levels were related to their use of violence (*r* = 0.322, *p* < 0.01).

### 3.3. Factors Contribution to Teachers’ Stress

In Table 3, intercorrelations between all study variables are presented. In a hierarchical multiple regression analysis, we correlated potential contributing factors with teachers’ stress. In model 1, we entered the control variables age, gender, and family size, and in model 2, we added time spent at work, class size, negative school climate, and teaching difficulties, as well as feelings of pressure. The results are presented in Table 4. The overall model explained 17% of the variability of teachers’ stress. Feelings of pressure (β = 0.320), negative school climate and teaching difficulties (β = 0.166), and time spent at work (β *=* 0.145) were all related to teachers’ stress.

## 4. Discussion

In the current study, we investigated teachers’ personal and school-related factors that may contribute to elevated levels of stress. Our results indicated that a majority of teachers (60.2%) experienced elevated and high levels of stress, and that higher levels of stress were associated with the use of violence. Furthermore, work-related stress and student-related stress were associated with the use of violence (see Table 3). Many teachers reported feelings of pressure and experienced teaching difficulties (see Table 1).

In line with our hypothesis, the perceived feelings of pressure at work, negative school climate, and teaching difficulties were significantly associated with teachers’ stress. Thus, teacher-related school environment factors contributed to teachers’ stress. Our findings converge with previous studies that discussed how stressful working conditions in Sub-Saharan African and Asian schools resulted in increased teachers’ stress [5,7,23]. Increased stress levels among teachers have been previously documented [3,4,5,8,12,14,15,16,19,20,21,22,23]. Furthermore, high levels of stress have the potential to result in negative outcomes. For instance, stressed teachers may decide to leave the teaching profession, resulting in disruptions in the education sector. About 2500 secondary school teachers (4%) in Uganda left the teaching vocation for diverse reasons [13].

Our results shed light on the diverse sources of stress teachers experience that put them under pressure during the execution of their tasks. Most of the time, these are external stressors, such as the implementation of a new curriculum at lower secondary schools without the adequate training of the few available teachers, and little appreciation of the services of dedicated teachers by the public. Teachers’ struggles and concerns with their work and life balance may affect their class duties in a resource-limited education sector [6]. Our findings, in part, explain how teacher stress affects their behavior and performance, for instance, the association between teachers’ conditions of service, perceived stress, and the application of violence [8]. School-related conditions have the potential of increasing teachers’ stress that altogether increase the risk of the application of violence against students. Stressed teachers are more likely to use violence indiscriminately, regardless of the magnitude of the student misbehavior.

Furthermore, in the present study, feelings of pressure were greater for male teachers than for female teachers. This may be explained to some extent by the sample characteristics. The present study sample consisted of mainly male teachers (over 75%). In Uganda, male teachers have the responsibility of taking care of their families. In many cases, they had to engage in supplementary income-generating activities, like owning businesses, that have the potential of compromising their teaching duties.

Our findings were not in line with other studies that reported associations between teachers’ stress and demographic variables, such as age, family size, and socioeconomic status [9,20,21,22]. More so, other personal factors, including teaching experience and academic qualifications in this study, were not related to stress, a finding that was also not in line with previous findings [23]. This may be explained at least partially by the fact that teachers in Uganda plausibly have choices when it comes to their teaching work. Teachers who are dissatisfied with the much-cherished teaching profession may quit for other, better career options [13]. Furthermore, teachers may adopt strategies that help them manage teaching difficulties, for instance, the use of violence in the case of known difficult classes, like students in the ninth year of formal education [15]. With large classes that contain over 60 students in a classroom, teachers tend to focus on the few brilliant students who may be first at grasping the material, while disregarding the educational needs of other students. Furthermore, some teachers can act as mentors for their students, while other teachers may also use avoidance strategies, including the use of limited creativity in the teaching because of a lack of resources [6]. These strategies enable teachers to experience less stress as they perform their tasks but may also explain the poor teaching quality in Ugandan schools.

Our findings indicate that demanding working conditions are related to increased stress among teachers and the subsequent use of violent discipline. Therefore, teacher training institutions need to educate prospective teachers on the realities of the teaching profession. Teacher trainees need to be updated on the pros and cons of the teaching profession. Trainees need to be adequately equipped with skills for managing stress, self-care, multitasking, and classroom management techniques. In so doing, trainees will be ready to work effectively in a challenging work environment that is devoid of the necessary resources during their practicum and after they have qualified as professional teachers.

In examining the personal and work-related factors that are related to increased teachers’ stress levels, our study provides a basis for targeted interventions that can, in the short-term, reduce teacher stress, while, in the long-term, decrease the use of violent discipline [7,8]. Our results imply that we need to focus on interventions that directly address factors that result in heightened teacher stress. For instance, improving the working condition of teachers can include employing more teachers to handle large classes and reducing the time teachers spend at school [16]. The assumption here is that changes in the educational sector that focus on improving teachers’ working conditions have the potential to reduce stress. Additionally, there is a need for training teachers in classroom management techniques and alternative disciplinary approaches to enable teachers to have the skills necessary for handling extreme student behavior and discipline. For example, interventions, such as Interaction Competencies with Children for Teachers (ICC-T), have proved their potential in improving teacher and student relations, providing teachers with non-violent disciplinary techniques while also changing and reducing the perceived workload [7,27].

Teachers’ stress can also be reduced by improved social support from supervisors and professional networks, appreciation of their contribution by society, and seeking their input when formulating policies that affect their work. Such strategies can easily be implemented at the interpersonal level and school level.

Our study investigated the associations between working conditions and increased stress levels among teachers. However, our findings need to be replicated in longitudinal studies with larger samples to get a better understanding of how diverse work-related factors result in increased stress. Likewise, further research should examine other factors that contribute to heightened stress, including personal factors, such as past violence exposure, mental health, and attitudes towards violence. With a better understanding of the factors that cause stress among teachers in the school setting, we can refine interventions that reduce stress.

Our study adds more knowledge to the relation between the school context and teachers’ stress. With a large representative sample of teachers from 12 public secondary schools from southwestern Uganda, our findings can be generalized to the public secondary schools in that region. The measures used in the study have been used in East Africa before. However, some limitations need to be noted. First, the cross-sectional study design does not allow for the establishment of causality. However, the results are consistent with prior studies that established relationships between work conditions and teachers’ stress. Secondly, the use of self-reports is prone to bias and social desirability. Furthermore, the fact that our model only predicted 17% of the variability of teachers’ stress levels implies that other factors may explain teachers’ stress that we did not include in this study.

## 5. Conclusions

Our results indicate that teachers’ perception of their working conditions, but not their personal characteristics, are related to teachers’ stress levels. As teachers’ stress levels are related to negative educational outcomes, more efforts need to be geared towards improving the working conditions of secondary school teachers as a way of reducing stress. Teachers also require training in stress management techniques, self-care, and alternative disciplinary strategies.

## Figures and Tables

**Table 1 ijerph-18-02315-t001:** Teachers’ burden and satisfaction of work (%).

Items	Not True	Somehow True	Certainly True
*School environment and teaching problems*			
I work productively with other teachers.	78.7	18.9	2.4
I establish achievable goals for myself.	69.0	25.8	5.2
I guide students toward intellectual and emotional growth.	80.8	17.5	1.7
Organizing the classroom for instruction is easy.	47.4	45.7	6.9
The physical environment of my classroom (e.g., equipment, number of students) helps my teaching.	56.0	37.5	6.5
I utilize a variety of teaching methods in the classroom.	71.4	26.5	2.1
I maintain appropriate classroom control.	71.2	27.8	1.0
I respect my students.	83.2	16.5	0.3
I insist upon students’ respect for others and their property.	81.8	15.8	2.4
I work effectively with superiors (e.g., headteacher, dean of students, head of department).	77.6	21.0	1.4
I feel competent in the subjects that I teach.	6.0	12.0	82.0
I become involved in the personal lives of students.	14.8	46.7	38.5
I have a personal philosophy of education.	51.7	40.0	8.3
I am accepted and respected by my professional peers.	70.8	26.1	3.1
*Feelings of pressure*			
Personal crises affect my life as a teacher.	17.6	47.9	34.5
Curriculum demands pressure me.	26.6	44.1	29.3
I consider leaving the teaching profession.	47.0	30.0	23.0
I feel under pressure in teaching.	45.5	37.9	16.6
I feel inadequate as a teacher.	66.7	18.9	14.4
Making long-term plans is difficult for me.	48.5	33.3	18.2
Teaching takes time away from my family.	19.9	45.4	34.7
Lack of public acknowledgment of teachers bothers me.	23.0	37.0	40.0
Lack of academic freedom concerns me.	27.1	39.9	33.0
Lack of public funding for the education sector concerns me.	10.0	25.9	64.1
I feel my salary is adequate.	21.3	21.0	57.7
I have difficulty being productive within the psychological climate of the school.	34.7	44.3	21.0
Personally managing disruptive students in class is difficult.	16.8	33.0	50.2

**Table 2 ijerph-18-02315-t002:** Teachers stress level (%).

	Stress Level		
How Often…	No	Elevated	High
Personal-related stress	23.7	50.5	25.8
…have you felt tired?	9.6	9.6	80.8
…have you been physically exhausted?	15.8	17.5	66.7
…have you been emotionally exhausted?	25.7	26.5	47.8
…have you thought “I can’t take it anymore”?	48.1	22.7	29.2
…have you felt worn-out?	38.1	20.7	41.2
…have you felt weak and susceptible to illness?	24.1	23.7	52.2
Work-related stress	41.0	45.9	13.1
…have you felt worn out at the end of the working day?	21.6	15.9	62.5
…have you been exhausted in the morning at the thought of another day at work?	49.1	23.4	27.5
…have you felt that every working hour is tiring for you?	56.4	22.0	21.6
…have you had enough energy for family and friends during leisure time?	12.4	17.2	70.4
…has your work been emotionally exhausting?	32.0	24.0	44.0
…has your work frustrated you?	46.9	20.3	32.8
…have you felt burnt out because of your work?	51.2	21.0	27.8
Student-related stress	48.6	35.2	16.2
…have you found it hard to work with students?	42.1	23.1	34.8
…has it drained your energy to work with students?	47.4	22.4	30.2
…have you found it frustrating to work with students?	54.0	17.8	28.2
…have you felt that you give more than you get back when you work with students?	31.3	13.7	55.0
…have you been tired of working with students?	48.8	21.0	30.2
…have you wondered how long you will be able to continue working with students?	42.6	17.2	40.2
Total stress level	39.8	47.1	13.1

**Table 3 ijerph-18-02315-t003:** Correlation among demographics, burnout, burden, and satisfaction of work.

Study Variables	1	2	3	4	5	6	7	8	9	10	11
1. Age											
2. Gender	−0.141 *										
3. Working hours	0.122 *	−0.115									
4. Class size	−0.092	−0.044	0.140 *								
5. Family size	0.361 **	0.099	0.088	−0.037							
6. Personal stress	0.087	0.006	0.107	−0.174 **	0.135 *						
7. Work-related stress	0.109	−0.041	0.096	−0.125 *	0.079	0.656 **					
8. Student-related stress	0.002	0.059	0.052	−0.069	0.095	0.559 **	0.627 **				
9. Teacher stress total	0.074	0.013	0.097	−0.142 *	0.118 *	0.856 **	0.881 **	0.851 **			
10. Feelings of pressure	0.055	−0.126 *	−0.076	0.009	0.067	0.244 **	0.282 **	0.219 **	0.287 **		
11. Teaching difficulties	−0.065	−0.054	0.055	−0.003	−0.057	0.002	0.131 *	0.092	0.088	−0.192 **	
12. Violence	−0.016	−0.016	−0.047	0.037	−0.034	0.280 **	0.277 **	0.272 **	0.322 **	0.074	0.050

Note: * *p* < 0.05, ** *p* < 0.01.

**Table 4 ijerph-18-02315-t004:** Hierarchical regression analysis for variables predicting teacher stress.

	Model 1					Model 2				
Variable	B	SE(B)	β	*t*	*p*	B	SE(B)	β	*t*	*p*
Age	0.044	0.119	0.024	0.369	0.713	0.006	0.111	0.003	0.056	0.955
Gender	1.151	2.258	0.032	0.510	0.611	3.012	2.131	0.083	1.414	0.159
Family size	0.728	0.374	0.128	1.947	0.053	0.561	0.350	0.098	1.604	0.110
Working hours						0.075	0.030	0.145	2.500	0.013
Class size						−0.055	0.019	−0.166	−2.890	0.004
Teaching difficulties						0.776	0.270	0.166	2.872	0.004
Feelings of pressure						1.110	0.201	0.320	5.512	<0.001
*R* ^2^	0.021					0.166				
Δ*R*^2^						0.145				

Note: SE = standard error.

## Data Availability

The data presented in this study are available on request from the corresponding author. The data are not publicly available due to sensitive nature of the data.

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
