# Peer review of "Job Perceptions Contribute to Stress among Secondary School Teachers in Southwestern Uganda"

_ijerph, 2021, doi:10.3390/ijerph18052315_

Round 1
Reviewer 1 Report
First of all, thank you for accepting the marked suggestions. From my point of view, the article has improved substantially in focus and coherence. Congratulations.
Reviewer 2 Report
This is a very well written paper.
I would only suggest to the authors to emphasize and interpret by a few sentences the main findings derived from Tables 3 and 4 (correlations and regression analysis) in the text followed by them.
This manuscript is a resubmission of an earlier submission. The following is a list of the peer review reports and author responses from that submission.
Round 1
Reviewer 1 Report
A pleasure to read. Well designed and reported research on such an important topic. Teasing out some of the statements that had a high percentage of Not True and Certainly True responses on Teachers' burden and satisfaction suggest that, in addition to the work environment, respect and emotional development are not commonly perceived as the teachers' role and responsibility. Importantly, however, these aspects in child development will likely have a carry-over positive impact on class room control, respect and productivity between teachers and other common burdens/dissatisfactions/stress levels experienced in the 'Not True' beliefs explored. Recommendations from this research could include a focus on respect within the education department, as well working conditions and non-violent disciplinary techniques.
Author Response
Reviewer #1:
- Recommendations from this research could include a focus on respect within the education department, as well working conditions and non-violent disciplinary techniques.
Response: Following this important suggestion, we have extended our discussion and implication sections (p. 8) and it now reads as follows:
In examining the personal and work-related factors that are related to increased teacher stress levels, our study provides a basis for targeted interventions that can in the short-term reduce teacher stress while in the long-term decrease the use of violent discipline [7-8]. Our results imply that we need to focus on interventions that directly address factors that result in heightened teacher stress. For instance, improving the working condition of teachers can include employing more teachers to handle large classes and reducing the time teachers spend at school [16]. The assumption here is that changes in the educational sector that focus on improving teachers’ working conditions have the potential to reducing stress. Additionally, there is a need for training teachers in classroom management techniques and alternative disciplinary approaches to enable teacher having skills necessary for handling extreme student behavior and discipline. For example, interventions such as Interaction Competencies with Children for Teachers (ICC-T) have proved their potential in improving teacher-student relations, providing teachers with non-violent disciplinary techniques while also changing reducing the perceived workload [7].
Besides, teacher stress can also be reduced by improved social support from supervisors and professional networks, appreciation of their contribution by society and seeking for their input when formulating policies that affect their work. Such strategies can easily be implemented at the interpersonal level and school level.
Reviewer 2 Report
- The title does not reflect the objectives or the contents
- Unclear research question(s).
- Lack of relevance or connections between the different parts of the study (title, objectives, background, etc)
- I am not sure this paper is within the scope of this Journal!
Author Response
- The title does not reflect the objectives or the contents
- Unclear research question(s).
- Lack of relevance or connections between the different parts of the study (title, objectives, background, etc)
Response: In the revised version of the manuscript we have re-written the theory section and changed the focus of our line of argument to clearly link the different sections of the manuscript in a logical and coherent way. These changes have been made throughout the revised manuscript.
- I am not sure this paper is within the scope of this Journal!
Response: The original manuscript was written for the IJERPH Special Issue of "School Climate, Bullying, and School Violence" but our manuscript was removed from this issue. Now we are refocusing on stress and not violence and currently our manuscript has been assigned to the IJERPH Health Behavior, Chronic Disease and Health Promotion section.
The focus on teacher stress is a global public health concern that requires attention of not only researchers but also educationalists.
Reviewer 3 Report
This article is based on previous studies and the purpose of this study is to know how the labor perceptions on the aspects that influence the stress of teachers in Uganda. It has a special influence on the influence of sociodemographic and school-related variables.
The argument of the article has an adequate theoretical and conceptual basis.
The investigated topic is of interest to the scientific community. The research has been properly designed and the methods are appropriate. The authors are congratulated for choosing an interesting topic for the educational context.
Based on the above, I would like to make a general assessment of the proposal (relevance, structure, method, discussion, strengths and weaknesses, etc.)
In the first place, it strikes me that the weight of violence in different parts of the text. While in the section of the objectives the violence initiates the speech, it does not appear in the title. It was necessary to make clearer if the focus of the research is to analyze the teachers' perceptions about what influences their stress or if the violence that occurs in Ugandan educational centers influences the stress of the teachers. This aspect should be reviewed by the authors to make a decision on which aspect is the one that directs the research.
On the proposed objectives: (1) Analyze whether work perceptions (feelings of pressure at work and the perceived school climate, teaching difficulties) contribute to teachers' stress. (2) Predict whether teacher stress levels will vary with sociodemographic and school-related variables.
They would be appropriate if the investigation is approached from stress and not from violence. If not, it would have to be reviewed. They address the investigated topic correctly and subsequently guide the presentation of the conclusions.
The conclusions are not supported by existing theory on the subject and previous research. They could be expanded although it is noted that the discussion has been extensive.
The research and practical implications are clearly identified in the article.
Regarding the references, indicate that since the theme is based on a current situation, it would be necessary to incorporate more updated references. Of 34 references, only 6 are from 2019 and 2020. It would be necessary to analyze whether this aspect influences the failure to find studies that support the results.
Author Response
Reviewer #3:
1) In the first place, it strikes me that the weight of violence in different parts of the text. While in the section of the objectives the violence initiates the speech, it does not appear in the title. It was necessary to make clearer if the focus of the research is to analyze the teachers' perceptions about what influences their stress or if the violence that occurs in Ugandan educational centers influences the stress of the teachers. This aspect should be reviewed by the authors to make a decision on which aspect is the one that directs the research.
2) On the proposed objectives: (1) Analyze whether work perceptions (feelings of pressure at work and the perceived school climate, teaching difficulties) contribute to teachers' stress. (2) Predict whether teacher stress levels will vary with sociodemographic and school-related variables.
They would be appropriate if the investigation is approached from stress and not from violence. If not, it would have to be reviewed. They address the investigated topic correctly and subsequently guide the presentation of the conclusions.
Response: Following the reviewer’s suggestions above, we have revised our manuscript to focus on teacher stress. This change can be noted in the revised introduction, and objectives among others. Additionally the sections on violence have been largely excluded from the write-up.
- The conclusions are not supported by existing theory on the subject and previous research. They could be expanded although it is noted that the discussion has been extensive.
Response: Thanks for this concerns. The conclusions are partly supported by previous studies. For instance the relationship between stress and personal characteristics. Mores so recommendation of improving teacher working conditions and in-service training have been recommended by previous research. We tried to make this clearer in our revision.
- Regarding the references, indicate that since the theme is based on a current situation, it would be necessary to incorporate more updated references. Of 34 references, only 6 are from 2019 and 2020. It would be necessary to analyze whether this aspect influences the failure to find studies that support the results.
Response: We added some new research references including Ramberg et al. (2020), Kabito and Wami (2020).